# Human Induced Pluripotent Stem Cell as a Disease Modeling and Drug Development Platform—A Cardiac Perspective

**DOI:** 10.3390/cells10123483

**Published:** 2021-12-09

**Authors:** Mohamed M. Bekhite, P. Christian Schulze

**Affiliations:** Department of Internal Medicine I, Division of Cardiology, University Hospital Jena, FSU, 07747 Jena, Germany; Christian.Schulze@med.uni-jena.de

**Keywords:** iPSC, cardiomyocytes, metabolic phenotype, genome-editing, disease modeling, drug testing

## Abstract

A comprehensive understanding of the pathophysiology and cellular responses to drugs in human heart disease is limited by species differences between humans and experimental animals. In addition, isolation of human cardiomyocytes (CMs) is complicated because cells obtained by biopsy do not proliferate to provide sufficient numbers of cells for preclinical studies in vitro. Interestingly, the discovery of human-induced pluripotent stem cell (hiPSC) has opened up the possibility of generating and studying heart disease in a culture dish. The combination of reprogramming and genome editing technologies to generate a broad spectrum of human heart diseases in vitro offers a great opportunity to elucidate gene function and mechanisms. However, to exploit the potential applications of hiPSC-derived-CMs for drug testing and studying adult-onset cardiac disease, a full functional characterization of maturation and metabolic traits is required. In this review, we focus on methods to reprogram somatic cells into hiPSC and the solutions for overcome immaturity of the hiPSC-derived-CMs to mimic the structure and physiological properties of the adult human CMs to accurately model disease and test drug safety. Finally, we discuss how to improve the culture, differentiation, and purification of CMs to obtain sufficient numbers of desired types of hiPSC-derived-CMs for disease modeling and drug development platform.

## 1. Introduction

Many cardiac diseases are characterized by functional and structural abnormalities that lead to myocyte death [1,2]. To this end, several in vivo and in vitro models have been established to investigate environmental factors for cardiomyopathies such as myocarditis, cardiotoxicity, and non-ischemic as well as genetic cardiomyopathies [3,4,5,6,7,8]. However, the lack of knowledge about the underlying causes and mechanisms of cardiomyopathy has been recognized. Therefore, a better understanding of the pathogenic mechanisms is crucial to capture the early stages of disease development and to develop sensitive and effective drugs with fewer side effects. Due to difficulties in obtaining human adult cardiomyocytes (CMs), primary cell lines (neonatal CMs and adult CMs) from rodent hearts [9,10], immortalized cell lines (H9C2) [11], ANT-T antigen [12], AT-1 cells [13], MC29 [14], HL -1 [15], and AC16 [16], as well as mice and rats have provided the model for basic research and pharmaceutical investigation of human heart disease [5,6]. However, primary cell lines and immortalized cell lines may have genetic and metabolic abnormalities due to their origin, resulting in misleading cellular responses to pathological stress. In addition, some cases of human genetic cardiomyopathies, such as the Tmem43 (Transmembrane protein 43) mutation in the nuclear lamina-associated protein genes, which causes arrhythmogenic cardiomyopathy (ACM) in several families, cannot be modeled with Tmem43 knock-in or knock-out mice [17,18,19]. Therefore, animal models can be misleading when studying human cardiomyopathies and they fail to translate mouse research findings to humans due to interspecies differences in physiological characteristics (i.e., heart size, heart rate, and ion channel contributions) [20]. In addition, animal models are still time-consuming and relatively expensive to create. In this complex scenario, many drugs that perform well in preclinical primary and immortalized cell lines and rodent studies fail in humans due to lack of safety and/or efficacy. To overcome these limitations, it is more attractive to use an appropriate human disease model in vitro. This has led to the creation of alternative models, such as CMs, which are derived from human-induced pluripotent cell (hiPSC) and have an unlimited source of human cardiac cells that can be used to study heart disease at the cellular level and test therapeutics compounds [21,22]. Further, hiPSC technology has shown great promise and offers the possibility of overcoming ethical and safety concerns that have arisen with the use of human embryonic stem cell (hESC) [23]. Importantly, the technology used to generate iPSC from diseased or healthy individuals carries the genome of an individual’s cells and can be differentiated into a wide range of individual cells from the three germ layers (ectoderm, mesoderm, and endoderm) [24]. For this reason, these human disease models in the dish hold great promise for studying disease progression, which is not possible with cell lines and animal models. Furthermore, the ability of hiPSC to reflect patient pathophysiology has revolutionized the way we study cardiomyopathy [25]. They provide a powerful in vitro model system for disease modeling for genetic studies, high-throughput drug screening, prediction of toxic effects of drugs, personalized therapy, and potentially in vivo cell replacement therapy [25,26,27]. Therefore, iPSC represent a successful platform for disease modeling and drug discovery, even more, accurate than other in vitro cell line cultures and in vivo animal models [27]. This review summarizes the advantages and limitations of hiPSC-derived-CMs as an alternative to animal models and primary cells in modeling cardiac cell types and pursuing novel therapeutic strategies.

## 2. Reprogramming Somatic Cells to iPSC

HiPSCs are derived from reprogrammed terminally differentiated mature cells by overexpressing a series of specific transcription factors: Oct4 (octamer-binding transcription factor 4), Sox2 (sex-determining region Y-box 2), Klf4 (Kruppel-like factor 4), and c-Myc (cellular myelocytomatosis oncogene) to convert somatic cells into pluripotent cells [21]. Takahashi and Yamanaka reported the first successful reprogramming iPSC in 2006 from mouse embryonic fibroblasts by using a retroviral vector system [21]. This result was confirmed in 2007 by using human cells [28]. Yamanaka and colleagues had successfully reprogrammed adult human fibroblasts using OCT4, SOX2, KLF4, c-MYC (collectively referred as OSKM) factors with a retroviral system into hiPSC [28,29,30]. Thomson and colleagues also used OCT4 and SOX2, whereas replacing the other two with Nanog and Lin28 using a lentiviral system method [31].

The OSKM transcription factors work together and lead to modulating the efficiency of reprogramming by inactivating the somatic program and activating the pluripotency program. c-Myc and Klf4 act as pioneer factors as they can induce two properties at the earliest stage of the reprogramming process: first, the opening and activation of condensed heterochromatin to form a euchromatin conformation; second, the binding to different sites in the genome and the recruitment of enzymes for histone modification (acetyltransferases and demethylases) [30]. Interestingly, it was found that exogenous c-Myc is not necessary for the reprogramming of mouse and human fibroblasts to iPSC as they express c-Myc and Klf4 [29]. However, the efficiency of reprogramming is much lower and requires more time [29,32]. Moreover, c-Myc has been found to induce apoptosis and senescence, and this effect can be abolished by Klf4 [33]. Upon completion of global chromatin remodeling, exogenous Oct4 and Sox2 are now able to positively activate the Oct4, Sox2, and Nanog genes, leading to a revival of the autoregulatory loop that triggers the pluripotency network and represses differentiation genes [34]. Oct3/4 is considered a key pluripotency factor and likely alters the cell fate of tumor cells to ESC [29]. It was found that Klf4 can be replaced with Klf2 or Klf5 and Sox2 can be replaced with Sox1 or Sox3 [35]. The expression of OSKM factors in reprogrammed mouse fibroblasts requires only one week, whereas reprogrammed human fibroblasts require two weeks. During this period, the reprogrammed cells proliferate slowly and exhibit an incomplete/intermediate reprogrammed pluripotent state [36,37]. Most somatic cells have been shown to allow reprogramming to iPSC, including fibroblasts, peripheral blood mononuclear cells (PBMCs) such as T cells and endothelial progenitor cells (EPCs), cultured hair follicle-derived keratinocytes, urine-derived renal tubule cells, intestinal cells, adipose stromal cells, and others cells (Figure 1) [38,39,40,41,42,43,44,45,46]. An important recent development in this field has been the ability to perform reprogramming on PBMC, which have become the most commonly used cells for hiPSC reprogramming because blood samples are easy to obtain compared to fibroblasts from a skin punch biopsy [38,39,41]. In addition, isolating PBMCs is safe for the donor and not as costly as performing a skin biopsy.

## 3. Delivery or Induction Methods of OSKM Factors

Several delivery methods have been developed to improve the efficiency of iPSC reprogramming and reduce the risk of genomic insertion and mutagenesis [47]. Therefore, non-integrative delivery methods have been developed to overcome safety issues. Delivery methods can be divided into two categories: integrative methods, which are technically simple and have high efficiency, and non-integrative methods, which are technically relatively difficult and have low efficiency. Integrative methods include viral vector-based methods (retrovirus, lentivirus, and inducible virus) and non-viral vector methods (plasmid or linear DNA and transposons). Non-integrative viruses include viral (adenovirus and Sendai virus) [48], and nonviral vectors (episomal DNA vectors, RNAs, human artificial chromosomes (HAC), proteins, and small-molecule compounds) [49,50,51,52]. Integrative systems have been criticized because their vectors are integrated into the host genome, resulting in difficult differentiation of iPSC and posing a risk of insertional mutagenesis during transfection [29]. Sendai virus methods remain the most popular among researchers because they are effective in fibroblasts and PBMC and do not enter the nucleus of host cells [38,48,53,54,55,56]. In addition, vectors based on the Sendai virus have a replication deficiency that is lost after temperature-shift treatment or about 10 passages of newly obtained iPSC [53].

To generate iPSCs that are free from viral contamination, and also non-integrative systems, considerable progress has been made by transfecting DNA into cells using liposomes or electroporation, such as plasmids, episomal plasmids [57,58,59,60,61,62], minicircle DNA [63,64], removable piggyBac transposon followed by Cre transfection [65] and non-DNA based methods such as synthetic mRNAs [66,67], miRNAs (miR302/367 cluster) [68,69] and recombinant proteins [49,70]. Although proteins or mRNAs can generate iPSC, the protocols are costly and technically challenging. However, this approach is considered less mutagenic than DNA-based approaches and non-integrative approaches. Recently, the CRISPR (Clustered Regular Interspaced Short Palindromic Repeats)-associated Cas9 nuclease (CRISPR/Cas9) system was developed and used to reprogram human skin fibroblasts into iPSC [71,72]. This technique uses a fusion protein of the enzymatically inactive form of Cas9 (dCas9) that retains a specific binding ability to DNA without causing DNA double-strand breaks (DSBs) [73]. Moreover, a fusion of dCas9 with transcriptional activators allowed activation of the desired DNA locus with high specificity and efficiency with minimal off-target activity [74,75].

In summary, integrative methods and viral vector-based methods are not suitable for clinical application due to the risk of insertional mutagenesis, and non-viral, non-integrative systems are more likely to be used for clinical application of hiPSC generated, which are maintained under feeder- and xeno-free culture conditions.

## 4. Derivation of CMs Subtypes

In the past decades, only primary CMs isolated from rodent hearts (1~5 day old rats or mice) and cardiac murine atrial myocytes (HL-1) cell line have been used as beating CMs for functional analyses of cardiomyopathies [10,15,76,77]. However, HL-1 cells and primary CMs have different electrophysiology and cannot recapitulate the entire pathophysiological profile of human heart disease [15,78]. Moreover, obtaining CMs from human hearts for functional analysis and genetic studies in patients is a highly invasive procedure and impossible in most cases. Moreover, the low proliferative capacity of CMs limits their long-term culture. The generation of patient-specific hiPSC-derived-CMs overcomes this problem and provides a promising cell source for understanding the pathological mechanism and developing new therapeutic agents.

Since inherited arrhythmias and cardiomyopathies affect different cardiac chambers, it is necessary to develop different protocols to generate atrial, ventricular, and pacemaker-like CMs subtypes. With the discovery of hiPSC, protocols for robust CMs differentiation from hiPSC have improved over the past decade [22,79,80,81]. The differentiation of iPSC into CMs is a complex and multistep process, so small variations at each step can lead to different results. Most protocols for differentiation of CMs use small molecule compounds and chemically defined media [82,83,84]. The small molecule glycogen synthase kinase (GSK)-3b inhibitor CHIR99021 is used to promote differentiation of iPSC into Brachyury-expressing mesodermal lineages, followed by treatment with the canonical Wnt inhibitor IWR-1, IWP2, XAV939, or Wnt-C59, which promotes differentiation of these mesodermal cells into the Nkx2.5 expressing cardiac progenitor cells [85,86]. Interestingly, recent studies have described that cardiac progenitor cells efficiently generate atrial-like (myosin light chain 2 atrial isoform, Mlc2a) in the presence of retinoic acid and ventricular-like (myosin light chain 2 ventricular isoform, Mlc2v) CMs in the absence of retinoic acid [87,88,89]. Generation of sinoatrial node (SAN) pacemaker cells (hyperpolarization-activated cyclic nucleotide channel 4, Hcn4) from hiPSC was achieved through activation of BMP and retinoic acid signaling pathways as well as overexpression of TBX3 [90,91]. Moreover, inhibition of Neregulin1β/ErbB signaling can lead to a three-fold increase in SAN pacemaker cell differentiation from hESCs [92]. In these protocols, it was found that various recombinant growth factors, e.g., BMP4, can also be used to increase the efficiency of iPSC differentiation in CMs [85,91,93].

All of these CMs subtypes have well-defined electrophysiological properties and gene expression profiles [94] as studied by patch-clamp technology [95], single-cell RNA sequencing [96], and voltage and calcium imaging [97]. In addition, many protocols have been established to differentiate the iPSC into nonmyocyte cell populations of the heart, such as epicardial cells [98], cardiac progenitor cells [99], cardiac fibroblasts [100,101], and valve interstitial cells [102]. Further, to generate vascular smooth muscle cells (vSMCs) or endothelial cells (ECs) as vascular cell types, vascular mesodermal progenitors lineages were first induced from iPSC, then by treatment with platelet-derived growth factor (PDGF)-BB, activin A and transforming growth factor (TGF)-b1 to induce lineage-specific vSMCs [103,104] with vascular endothelial growth factor (VEGF)-A to induce ECs [104,105].

Despite these advances in differentiating CMs with currently available protocols, it is desirable to obtain pure CMs to improve the efficiency and clinical compatibility. One strategy is to use fluorescence-activated cell sorting or magnetically activated cell sorting to separate CMs from other cells [106,107]. An ideal approach is to use lentiviral vectors to introduce antibiotic resistance genes driven by CM-specific promoters in hPSCs, which resulted in 96% pure CMs [108,109]. Another important step forward was the development of a protocol to purify hPSCs-CMs to 99% purity using glucose depletion and lactate supplementation [110]. However, obtaining a pure population of cell subtypes is a challenge that remains to be solved [92]. These improvements will be of great benefit not only for disease model research and drug development and screening but also for potential clinical application approaches.

## 5. Challenges in iPSC-Derived-CMs Maturation

To achieve more accurate disease modeling and maximize the potential applications of hiPSC-derived-CMs for drug testing, characterization of the maturation features of CMs is critical. Some features can be used to estimate the maturation of CMs, such as cell shape, sarcomere, composition of myofibrillar isoforms, T-tubules, mitochondria, metabolic substrate, number of nuclei, electrophysiological properties, calcium handling, contractile force, gap junction distribution, and response to β-adrenergic stimulation (Table 1) [111,112,113,114,115].

It is known that human CMs require years to reach their adult phenotype in vivo [116,117,118]. Beating CMs can be generated from hiPSC within 6–8 days after differentiation [79,84]. However, iPSC-derived-CMs still have a circular/irregular shape that resembles the structure of immature human CMs, rather than the shape of rods like mature CMs [113,119]. In addition, these cells have a cell surface area 480 ± 32 µm2 compared to mature CMs, which have a cell area of about 1716 ± 150 µm2 and a diameter of 15–30 µm [120]. Furthermore, mature CMs were shown to have a sarcomere length of about ≈1.81–2.3 µm compared to ≈1.65 ± 0.02 µm in immature CMs [121,122], and the structure of sarcomeres in mature CMs is highly organized, and the Z-bands run parallel to the intercalation disk [115,123]. Further, the gap junction proteins connexin 43 (Cx43) and N-cadherin are increasingly concentrated in the intercalation disks at the ends of the mature CM, whereas they are circumferentially distributed in the immature CMs (Figure 2) [124]. Therefore, assessment of myofibrillar isoforms of proteins such as cardiac troponin I (ssTnI in immature CMs and cTnI in mature CMs) [125,126], titin (N2BA in immature CMs and N2B in mature CMs), and the ratio of myosin heavy chain (MHC) (β-MHC >> α-MHC in mature CMs) may provide useful information about the maturation of hiPSC-derived-CMs [127]. Another factor that can assess CMs maturation is the increase in sarcoplasmic/endoplasmic reticulum Ca^2+^ ATPase (SERCA) and the ratio of β1 to β2 adrenergic receptors (≈70 β1:30 β2) in mature CMs [128,129,130]. In addition, membrane potential in ventricular immature CMs (≈ −50 to −60 mV) compared to ventricular mature CMs (≈ −90 mV) [115,123,131]. Moreover, the values of the Nav1.5 sodium channel increase in mature CMs for upstroke velocity (≈44 to 50 V/s in immature CMs and ≈188.7 to 250V/s in mature CMs) [121,131,132]. In adult CMs, the sarcolemmal voltage gated L-type Ca^2+^ channels are located at the t-tubule network close to the sarcoplasmic reticulum (SR), which leads to synchronize release of Ca^2+^ from SR via ryanodine receptors (RyRs) in a rapid way during an action potential [133]. While undeveloped SR has been reported in immature hiPSC-derived-CMs, that is, express lower SERCA and cardiac SR luminal auxiliary proteins (calsequestrin, junctin, and triadin), which form a protein complex associated with RyR2 [134]. Also, SR is mainly distributed in the perinuclear region [135,136]. Furthermore, immature hiPSC-derived-CMs lack t-tubule network [137,138]. This results in poor co-localization of Ca^2+^ channels and RyRs, which contribute to unsynchronized and slow kinetics of the Ca^2+^ transients, which contribute to unsynchronized and slow kinetics of the Ca^2+^ transients [135,139].

## 6. Metabolic Characterization of hiPSC-Derived-CMs

The major sources of ATP production in the normal adult heart are fatty acids (FAs), glucose, pyruvate, amino acids, lactate, and ketone bodies [140]. It is known that under resting conditions, the adult heart generates ATP via mitochondrial oxidative phosphorylation, of which ≈70% is derived from FAs by β-oxidation [141]. The flexibility to switch between FA oxidation and glucose oxidation is essential for normal physiological cardiac function [142]. Mitochondria occupy ≈20% to 40% of the volume of the adult myocyte and are densely packed throughout the cell along the myofibrils. However, mitochondria in non-differentiated hiPSC are arranged perinuclearly, which is considered a “stem cell” property [113,143,144]. Further, immature hiPSC-derived-CMs depend mainly on glycolysis for energy production, and mitochondria are round and less fused into a network structure, also the inner membrane cristae are less mature [113,145,146]. In contrast, mature hiPSC-derived-CMs contained mainly elongated mitochondria and more mature cristae, accompanied by dramatic changes in the dimension of the mitochondrial network and a shift to β-oxidation for ATP synthesis as in normal adult hearts (Figure 3) [113,147]. Interestingly, several studies have shown that switching metabolism from glycolysis to fatty-acid β-oxidation in CMs promotes cellular maturation [148,149,150]. These observations suggest that the high activity of oxidative phosphorylation correlates with mitochondrial elongation and is consistent with the hypothesis that elongated mitochondria and their networks are more efficient at producing energy and can distribute energy over long distances [113,151]. In addition, mitochondrial maturity has been found to correlate with energy metabolism and ATP demand of mature hiPSC-derived-CMs. These important differences should be taken into account when using hiPSC-derived-CMs. Further studies on the maturation of iPSC-derived-CMs are needed in the future to develop optimal methods for more efficient differentiation of CMs that have the typical structure and physiological properties of adult CMs.

## 7. Applications of Genome Editing Technology in hiPSC

Recently, the application of genome editing in hiPSC has been widely associated. Their combination can further enhance the power of hiPSC in pathophysiology study, drug screening, and new drug development [152,153]. Recent genome editing techniques can be divided into three main tools: zinc finger nucleases (ZFNs) [154,155], transcription activator-like effector nucleases (TALENs) [156] and CRISPR/Cas9 can be used to exchange relevant nucleotides at specific positions [157,158]. The three genome editing techniques that are used have a similar mechanism and are based on an endonuclease activity that specifically makes DSBs in DNA at desired locations in the genome. The DSB triggers DNA repair via two different pathways, homology-directed repair (HDR) or non-homologous end joining (NHEJ) [159,160,161]. In contrast to ZFNs and TALENs, CRISPR/Cas9 targets a specific genomic site using the protospacer motif (PAM) and single-guide RNA (sgRNA or gRNA) to induce a DSB complementary to the CRISPR sequence [159,162]. Furthermore, modifications to the catalytic site of the Cas9 nuclease enabled the generation of dead Cas9 (dCas9) fused to a deaminase (e.g., adenine or cytidine), allowing the conversion of guanine-cytosine pairs to adenine-thymine or vice versa, which enabled the development of single base editing without a DSB [74,163,164]. In this regard, CRISPR-Cas9 has become the standard technology for precise genome manipulation by providing genome editing, epigenetic modulation, and transcriptional control.

Analysis of disease-specific hiPSC-derived-CMs compared to wild-type hiPSC-derived-CMs prepared from healthy donor cells as controls can lead to unreliable results because the genetic and epigenetic background of the cells is largely unknown [165,166]. Even when control cells from non-diseased individuals in the same family (first-degree relatives) are used, the patients from whom the disease cells are derived share only ≈50% of the genome [166,167,168]. The importance of having an isogenic cell line increases the chance of identifying the role of a particular mutation in the disease and circumventing heterogeneity between patients [169]. CRISPR/Cas9 technology could serve as a tool to correct pathogenic gene mutations in disease-specific hiPSC and can also be applied to wild-type hiPSC to generate the specific mutation [170]. Thus, this technology can be used to generate isogenic control iPSC so that disease-specific hiPSC-derived-CMs and control iPSC-derived-CMs share the same genetic background and differ only at the mutation site to allow more accurate disease modeling [171,172].

## 8. Strategies for Generating hiPSC for Heart Disease Modeling

Methods for modeling hiPSC diseases generally rely on either hiPSC reprogrammed from patient-derived cells from mutation carriers or genetically engineered wild-type hiPSC using genome editing approaches such as CRISPR/Cas9, ZFN, and TALEN [152,171,172,173,174,175]. In addition to genome editing approaches, iPSCs or the differentiated CMs can be genetically modified by reducing the expression of specific mutant proteins, e.g., by RNA interference (Figure 4) [176].

One of the major causes of heart disease is various genetic mutations. However, different mutations in the same gene may have different mechanisms at the molecular level but lead to similar clinical outcomes of heart disease [177,178]. Advances in patient-specific hiPSC-derived-CMs research have provided a platform to effectively study patient- and disease-specific heart disease in vitro [179,180,181]. Moreover, these cells have recapitulated cellular electrophysiological changes in the heart of patients [182,183]. The use of patient-specific hiPSC-derived-CMs may be useful for basic science investigations as well as for patient-specific therapeutic screening and personalization of therapy. In 2010, Carvajal-Vergara and coworkers obtained iPSC carrying the heterozygous mutation of the PTPN11 gene from patients with leopard syndrome, which is often associated with severe hypertrophic cardiomyopathy (HCM) [184,185]. Interestingly, these iPSC-derived-CMs showed abnormal size and nuclear localization [184]. Since then, many different iPSC lines have been generated. In addition to modeling genetic cardiomyopathies, iPSC-derived-CMs have also been used to model environmental factors that cause cardiomyopathies, such as the cardiotoxicity of anticancer drugs [186], nonischemic cardiomyopathies [187], peripartum [188], and diabetic cardiomyopathy [189,190,191,192], or even infection with Coxsackie B3 virus [193] or parasitic flagellate protozoa [194].

Looking at the literature in previous years, the most common genetic heart disease analyzed is the various types of Long QT Syndrome (LQTS) mutations [195,196,197,198,199,200]. LQTS is one of the most common types of cardiac channelopathies and is due to delayed repolarization in the ventricular phase, which can lead to sudden cardiac death [201]. The hiPSC-derived-CMs carrying this mutation were used for functional analysis and showed a prolonged QT interval [202]. Patient-derived iPSC-derived-CMs carrying mutations of the alpha subunit of the voltage-gated potassium channel subfamily Q member 1 (KCNQ1; also known as KvLQT1 or Kv7.1) showed impaired membrane trafficking of slow IKs channels and cause LQT1 [203,204]. Itzhaki et al. obtained LQT2 hiPSC-derived-CMs carrying a mutation in the alpha subunit of the voltage-gated potassium channel subfamily H, member 2 (KCNH2), which encodes the α-subunit of the Kv11.1 channel (human ether-a-go-go-related gene; hERG) responsible for conducting the fast delayed rectifier potassium current (IKr) in CMs [205]. In addition, in vitro models of disease-specific hiPSC-derived-CMs were generated from patients with Timothy syndrome (LQT8), which leads to cardiac arrhythmias caused by a mutation in the subunit of the voltage-gated calcium channel alpha1 C (CACNA1C; responsible for the L-type calcium current, ICa,L) [206]. Interestingly, in another model, a cyclin-dependent kinase 5 (CDK5) was identified as a regulator of L-type Ca^2+^ channel function. Inhibition of CDK5 with roscovitine was able to rescue the phenotypes in iPSC-derived-CMs from Timothy syndrome patients [207].

In addition, other disease-specific hiPSC-derived-CMs have also been produced for inherited arrhythmias, including various types of LQTS mutations, including, potassium channel inwardly rectifying channel subfamily J member 2 (KCNJ2) [208], calmodulin 1 (CALM1), or calmodulin 2 (CALM2) [209,210], sodium channel protein type 5 subunit alpha (SCN5A; Brugada syndrome type 1) [211,212]. Another channelopathy that has been used to generate a disease model is catecholaminergic polymorphic ventricular tachycardia (CPVT). Sasaki et al. generated CMs from CPVT patient-derived iPSC and identified S107 as a potential therapeutic agent because preincubation with S107 resulted in a reduction in isoprenaline-induced delayed after depolarizations [213].

Several other studies have generated iPSCs-derived-CMs models associated with HCM. HCM is characterized by hypertrophy of the ventricular walls and/or septum, resulting in decreased cardiac output [214]. Many studies have examined CMs derived from HCM patient iPSC with mutations in genes encoding sarcomeric proteins or other mutations identified in patients with HCM, such as ACTC1 (cardiac actin) [215], ALPK3 (α-kinase 3) [216], BRAF (B-Raf proto-oncogene, serine/threonine kinase) [217], FBN1 (Fibrillin 1; Marfan syndrome) [218], FXN (Frataxin; Friedreich ataxia) [219], GLA (Galactosidase α; Fabry disease) [220]. MT-RNR2 (Mitochondrially encoded 16S rRNA) [221], MYBPC3 (Myosin binding protein C3) [222,223,224,225,226,227,228,229,230,231], MYH7 (β-myosin heavy chain) [232,233,234,235,236,237], MYL2 (Myosin light chain 2) [238], MYL3 (Myosin light chain 3) [239], PRGAG2 (Protein kinase AMP-activated non-catalytic subunit gamma 2; Wolff-Parkinson-White Syndrome) [240,241], PTPN11 (Protein tyrosine phosphatase non-receptor type 11) [184,242]. RAF1 (Noonan syndrome) [242], SCO2 (Cytochrome c oxidase assembly protein) [243], TNNT2 (Cardiac troponin T) [222,244], and TPM1 (Tropomyosin-1) [228,229]. Interestingly, these mutant hiPSCs-derived-CMs were larger and exhibited a higher frequency of sarcomere disorganization. These cells also recapitulated abnormal calcium handling and exhibited abnormal electrophysiological and/or contraction measurements [245].

In addition to inherited cardiac arrhythmias, dilated cardiomyopathy (DCM) is one of the most important forms of cardiomyopathy [246]. DCM is manifested by enlargement of the left ventricle in combination with decreased ventricular thickness and heart failure due to severely impaired ejection fraction [247,248]. Based on family history and clinical findings, previous clinical studies have suggested that the most common mutated gene in familial DCM cases is Titin [249], followed by LMNA (lamin A/C) [250], MYH7 and MYH6 [251], SCN5A [252], MYBPC3 (Myosin-binding protein c, cardiac type) [253], and TNNT2 [254]. Interestingly, several known hiPSC-derived-CMs have been established to study inherited DCM, including desmin (DES) [255], function-related protein (FKRP) [256], LMNA [257,258,259,260,261], phospholamban (PLN) [262,263,264,265], RNA binding motif protein 20 (RBM20) [266,267], TNNT2 [128,244,262,268,269], Titin [269,270,271]. Patient-specific hiPSCs were produced from diseased members and CMs were generated and examined. After prolonged culture, the resulting CMs showed sarcomere disarray, defects in Ca^2+^ handling, and decreased contractility [268,272]. Sarcomere disarray was ameliorated by the addition of metoprolol and β-adrenergic blocker. In addition, decreased contractility could be enhanced by adenoviral overexpression of SERCA2a [268].

In addition, Sato Y et al., previously developed a model for late-onset of Pompe disease-hiPSC, an autosomal inherited metabolic disorder caused by a mutation in acid alpha-glucosidase (GAA). Interestingly, patient-specific hiPSC-derived-CMs exhibited higher levels of glycogen, mitochondrial dysfunction, and disordered CM fibers observed in the typical features of Pompe disease cardiomyopathy [273]. Although numerous studies have investigated the pathophysiology of heart disease using patient-specific hiPSC-derived-CMs, it is still challenging to fully recapitulate the disease phenotype, as the use of hiPSC-derived-CMs can lead to controversial results, especially when studying late-onset heart disease due to the immaturity of hiPSC-derived-CMs for disease modeling [274,275,276].

## 9. Consideration of hiPSC for Use in Modeling Heart Disease

Despite the extensive advantages of hiPSC-derived-CMs model as a platform for heart disease research and development of new therapeutic agents, however, there are still unresolved issues related to the maturation of hiPSC-derived-CMs that pose an obstacle to adult cardiovascular disease modeling. To overcome these limitations, several studies have contributed to the development of protocols to improve the maturation of hiPSC-derived-CMs (Figure 4). One method is to extend the duration of hiPSC-derived-CMs culture to more than 4 weeks [113]. In addition, culturing iPSC-derived-CMs for 3–7 days in a fatty acid-based medium improved the maturation of hiPSC-derived-CMs and showed significant changes in various aspects such as cell size, cell morphology, sarcomere length, gene and protein expression, metabolic changes, and percentage of multinucleate CMs, as well as an increase in calcium release and reuptake rates [113,277,278]. In addition, many laboratories are currently attempting to accelerate the maturation of CMs by treating CMs with triiodothyronine (T3) or dexamethasone (DEX), electrical or mechanical impulses, and culturing CMs in three-dimensional (3D) systems without/with scaffold materials [279,280,281,282,283,284,285]. However, the human native myocardium exhibits additional structural and functional complexity in terms of complex cell-extracellular matrix (ECM), cell–cell, and tissue-level interactions that are poorly represented by 2D cellular in vitro models. In this context, the recapitulation of native cardiac tissue structures and physiological function in vitro can be achieved by using 3D cellular constructs as new-generation models, which could improve the recapitulation of cardiac physiology and pathophysiology. Various types of 3D constructs have been produced by combining hiPSC-derived-CMs with non-cardiac cell types found in the heart, such as cardiac fibroblasts, endothelial cells, smooth muscle cells, and immune cells, in ratios similar to those found in the human heart [286,287]. 3D constructs of the human heart have been made with different compositions and architectures to reproduce cardiac spheroids (using non-adhesive U-shaped wells, hanging drops, or stirred cultures), artificial heart tissue, organoids, and heart-on-chip models [288,289,290]. However, organoid models of heart cultures still lack the vascularization that supplies oxygen and nutrients to the muscle [291]. The lack of oxygen and nutrients leads to the formation of a necrotic core [291]. However, the ever-growing field of organoid or gastruloids and artificial cardiac tissue technology, which offers a considerable number of features through the formation of small size organoids or the development of vascularization to solve this problem, will lead to a better understanding of disease pathogenesis at the tissue and organ level [288,292,293]. Recently, human cardiac organoids were used for modeling most of the cardiovascular diseases, which mimic the tissue architecture of heart, including cardiomyocytes, endothelial, stromal cells, and epicardial cells in vivo [122,294,295,296]. Further, many studies focus on vascularization of the cardiac organoids to overcome the lack of oxygen and nutrient diffusion, by coating the organoids with an extracellular matrix containing endothelial cells [297], cultured cardiac organoids with hPSC-derived endothelial cells [298], and fusion of two subregional organoids to form a complete organ [299]. Moreover, cardiac tissue engineering field has rapidly evolved over the past decade and offers a biomedicine discipline attempting to combine scaffolding polymers with cardiovascular cells to create cardiac tissue-like structures for drug screening, disease modeling, and cardiac repair [300,301,302,303,304,305]. Interestingly, the use of heart-on-a-chip models has recently increased, providing a controllable tool, mean oxygen delivery [306,307,308,309], pH [308], shear stress with a certain flow rate of the culture medium [310,311], temperature control, and electrical or mechanical stimulation, as well as continuous monitoring and measurement of parameters of the physiological responses of the cells. Unlike cardiac spheroids, cardiac patches can be seeded into a scaffold to form a layer (4 × 4 cm) of CMs or together with other cells without losing functional properties [312,313]. However, cell sheet layering methods still have some limitations, mainly related to the number of cell layers before necrosis of the grafted patch occurs after its implantation. By using hiPSC-derived-CMs and endothelial cells encapsulated in tissue-based hydrogels using a 3D printing method, vascularized, thick (2–7 mm) cardiac patches could be fabricated that exhibit cell viability and contractility [314].

However, despite the promising results of preclinical research, further studies are needed to develop effective and more efficient strategies for cardiac regeneration in vivo. Finally, it should be considered that the use of hiPSC-derived-CMs could provide basic information about the pathology and pathophysiology of the heart, and thus is valuable for understanding the inherited disease, efficient drug testing, and the drug development process.

## 10. Drug Testing and Drugs Development Using hiPSC-Derived-CMs

Currently, the development of new drugs is expensive and requires many years of preclinical research with in vitro and in vivo testing as well as human clinical trials before they are allowed to reach the market [315]. In most cases, only a small fraction of safe drug candidates reach the market after years of multiple processes [316,317]. Many drugs have been withdrawn from the market despite passing in vivo and clinical tests. An example of this is the increased risk of QT prolongation of heart rate and ventricular arrhythmias, which was a factor in one-third of drugs withdrawn from the market due to safety concerns [318,319,320,321]. This may be due to differences between experimental animals and humans, such as the expression profile of ion channels and the physiological properties of heart rate (~10-fold in mice at rest), which limit the utility of mice to study the effects of antiarrhythmic drugs [322]. As a result, researchers often turn to immortalized cell lines such as HL-1 cells due to the difficulties in obtaining and maintaining human primary cardiac cells. These challenges significantly slow the progress in drug development. Recently, researchers have turned to hiPSC-derived-CMs as an excellent alternative and a more reliable tool in the earliest stages of drug development to detect side effects of a new potential therapeutic agent that may cause prolongation of the QT interval before it is used in animal studies and clinical trials [323]. In addition, hiPSC allows us to study the behavior of ion channels in real CMs rather than in cell lines that are not myocytes and overexpress the potassium channel because of the response to compounds tested with hiPSC-derived-CMs in vitro is similar to that of the human body [324]. The effects of new drugs can be assessed in several ways, including electrophysiological responses using microelectrode arrays (MEAs), patch-clamp, or Ca^2+^ oscillation measurements. Their thorough analysis allows determining whether a particular drug blocks or activates one of the ion channels involved in action potential generation [325,326,327]. In addition, hiPSC-derived-CMs provide a higher safety model for preclinical testing to verify the cardiotoxicity of the human-specific response, which is a common reason for withdrawal of drugs from marketing or rejection in the final stages of clinical trials. Blockade of the ion channel encoded by the hERG has been associated with prolongation of the QT interval, which in turn is correlated with cardiac arrest and death from torsades de pointes (TdP) [328]. The study of hERG activity using patch-clamp currents in the absence and presence of a drug is now commonly part of safety pharmacological risk assessment [329]. In previous study, iPSC-derived-CMs were used to test 31 inotropic and 20 non-inotropic compounds previously characterized for their effects on contraction changes in CMs and were found to have a specificity of 70% and a sensitivity of 87% [330]. This shows that iPSC-derived-CMs can be used for cardiotoxicity screening [331]. Another study by researchers at the US Food and Drug Administration confirmed the potential use of iPSC-derived-CMs for predicting drug risk for cardiac arrhythmias [332]. Because iPSC-derived-CMs have a significant advantage as a system for testing potential drug candidates, they are beginning to be used in drug discovery research. For example, to overcome the high toxicity of doxorubicin, hiPSC-derived-CMs have been used to test a liposomal formulation of doxorubicin that resulted in little or no uptake into human CMs and prevented the signs of cardiotoxicity. These results support the clinical decision to move into Phase I clinical trials with the drug formulation [333].

HiPSC-derived-CMs can also serve as models for potential drugs development against heart disease (see the review by Hnatiuk et al.) [334]. Interestingly, Leonid Maizels and co-workers investigated the drug response in a disease-specific hiPSCs-derived-CMs model of the autosomal recessive form of catecholaminergic polymorphic ventricular tachycardia type 2 (CPVT2) (because of the D307H-CASQ2 mutation). The authors found that flecainide application has protective against the development of arrhythmias in the CPVT2-hiPSC-derived-CMs, which correlates with clinical data collected from the same cell donor patient [335]. In a recent study by Knottnerus et al., accumulation of long-chain fatty acid oxidation intermediates was found in iPSC-derived-CMs from patients with very long-chain acyl-CoA dehydrogenase deficiency (VLCADD), who present life-threatening arrhythmias, leading to cardiac arrhythmias. This study proved that the use of agents such as etomoxir and resveratrol, resulted in reversion of these abnormalities and restoration of the correct phenotype, suggesting the possible use of these compounds in the therapy of VLCADD-CMs [336]. Moreover, as demonstrated by Loboda and Dulak, hiPSC generated from Duchenne muscular dystrophy (DMD) patients and differentiated into CMs showed an increase in arrhythmic events rate in comparison to isogenic control. Interestingly, stimulating this CM with propranolol abolished this effect in vivo, suggesting it as a potential clinical treatment for patients developing DMD-associated cardiomyopathy [337].

The use of iPSC-derived-CMs in modeling of heart disease can significantly accelerate the drug development process, but without diminishing the importance of downstream characterization and validation in animal models.

## 11. Obtaining a Reproducible and Sufficient Number of hiPSC-Derived-CMs

In cardiovascular research, it is not possible to obtain a sufficient number of all cell types because of the high level of invasive procedures required to obtain them. In addition, the low proliferative capacity of CMs limits the ability of researchers to maintain these cells in culture. In contrast, hiPSC technology has triggered a paradigm shift in drug discovery and clinical trials. Furthermore, hiPSCs circumvent many of the problems associated with animal and primary cell models and enable the mass production of large numbers of patient- and disease-specific CMs and other cardiac cell types [338,339]. The main advantage of hiPSC-derived-CMs models is that they can expand indefinitely and provide more physiologically and clinically relevant, reproducible human cell models for in vitro disease modeling and high-throughput drug screening with low cost and a limited number of animals for experiments [182,340,341]. To enable high throughput, a multilayer flask format with a surface area of up to 875 cm^2^ has been developed (Corning, Falcon) [342]. In addition, several automated systems, such as multi-channel liquid handling robots on the bench, have been used to maintain hiPSC in a 96-well format, which helps to minimize technical and biological variance [343,344], an important innovation has been the development of platforms to differentiate hPSCs into CMs at 80% purity in suspension culture [345]. Despite these capabilities, there are still some limitations. Conventional monolayer cultures provide limited surface area and require repeated subculturing to obtain large numbers of hiPSC-derived-CMs. In addition, the replacement of 3D culture models for drug discovery and therapy will require large-scale production of hiPSCs-derived-CMs. Therefore, large-scale production of 3D CMs from iPSC can be achieved by various strategies, including suspension culture [346,347,348] or suspension using microcarrier substrates [349]. Further, to reduce the physical stress on cells caused by mechanical agitation or collisions with microcarriers, other researchers have developed bioreactor tanks and obtained 40 million ventricle-like CM in 100-mL bioreactors with purity up to 85% [350]. Interestingly, Konagaya et al., developed a compact automated culture system for maintaining hiPSC undifferentiated state by the system for 60 days [351].

## 12. Conclusions and Challenges

Previous models for drug screening in humans have been based on immortalized tumor-derived cell lines and animal models, which are generally expensive, have low throughput, and differ in human cardiac physiology. While primary human CMs directly model the effects of a drug on humans, their availability and expansion capacity are limited and finite compared to in vitro derived cell lines. The limited access to human cardiac tissue biopsies and primary CMs has promoted the idea of using hiPSC-derived-CMs not only for drug discovery but also for disease modeling. In this context, the combination of hiPSC-derived-CMs with genome editing techniques such as CRISPR/Cas9 has become an indispensable tool of human cellular models to mimic the phenotype of human diseases in vitro and further advance our understanding of specific genetic cardiomyopathies and drug development in cardiovascular medicine. Despite the great advantages and high efficiency of obtaining CMs from iPSC, many challenges still need to be overcome to use hiPSC-derived-CMs for drug discovery. This is especially true for the structure, metabolism, signal transduction, and maturation of hiPSC-derived-CMs. This is particularly important for studying the phenomenon of QT elongation, which is often a side effect of the drugs tested. Recently, several studies have attempted to improve the maturation of hiPSC-derived-CMs in different ways, including the use of T3 hormone, metabolic maturation, 3D construction, mechanical stress, electrical stimulation, long-term cultivation, and co-culture with other cell types such as cardiac fibroblasts and endothelial cells. The advancement of 3D cardiac structures enabled the development of systems more accurately, the physiological phenotype and drug-induced responses, in comparison to their 2D counterparts. Further, the automated cell culture results in large-scale production of hiPSC, which is useful to improve the reproducibility of cell cultures and reduce the time and effort of researchers. Nevertheless, there are still some limitations to be resolved. On the one hand, for CM transplantation, mature CMs cannot be transplanted into damaged myocardium, and neonatal cells can only survive to form new myocardium [352], so this may be disadvantageous for regenerative medicine applications. Therefore, the mature iPSC-derived-CMs may be not suitable for regenerative medicine. Second, it is the scalability of iPSC-derived-CMs production, which is inversely proportional to scalability, cost, and reproducibility. Therefore, a combination of ascorbic acid and growth factors can effectively improve the differentiation efficiency and maturity of iPSC-CMs [93]. However, the emerging iPSC technology cannot completely replace animal models but represents considerable advantages over classical in vitro models, by offering more clinically relevant tissue samples, using efficient, high-performance tools to test drugs efficiently, as well as drug development process, reducing time and cost-effects than those currently available.

## Figures and Tables

**Figure 1 cells-10-03483-f001:**
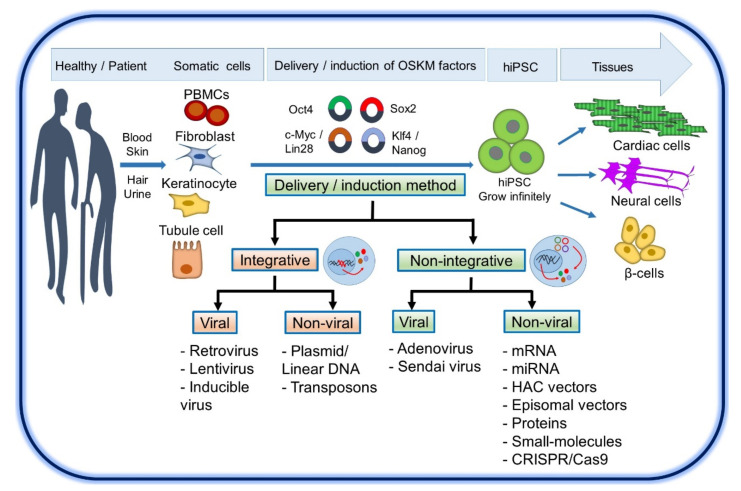
Schematic representation of delivery or induction of OSKM factors to convert somatic cells into pluripotent cells and differentiation the iPSC into a wide range of individual cells from the three germ layers.

**Figure 2 cells-10-03483-f002:**
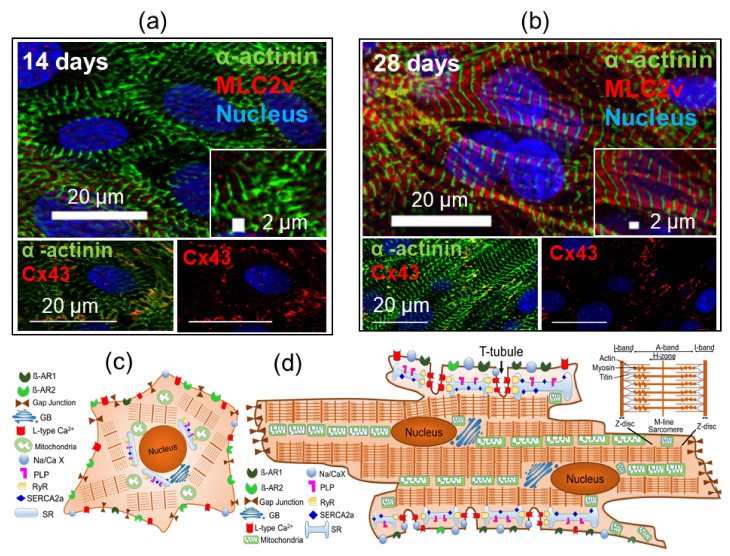
A comparison of immature and mature CMs. Representative image of immunofluorescent staining of cardiac marker α-actinin and MLC2v as well as gap junction proteins connexin 43 (Cx43) and nucleus (blue) at day 14 (**a**) and days 28 (**b**). Schematic drawing of the major difference between immature and mature CMs (**c**,**d**). Note that immature CM (**c**) differ from mature CM (**d**) with respect to shape, sarcomere structure and length, T-tubules, mitochondria, number of nuclei, sarcoplasmic area, gap junction Cx43, and β1-adrenergic expression. Abbreviation: sarcoplasmic reticulum, (SR); ryanodine receptor, (RyR) channels; sarco/endoplasmic reticulum Ca^2+^ ATPase, (SERCA); Golgi body, (GB); β-adrenergic receptor (β-AR); Na+-Ca^2+^ exchanger (Na/Ca X); phospholamban, (PLP).

**Figure 3 cells-10-03483-f003:**
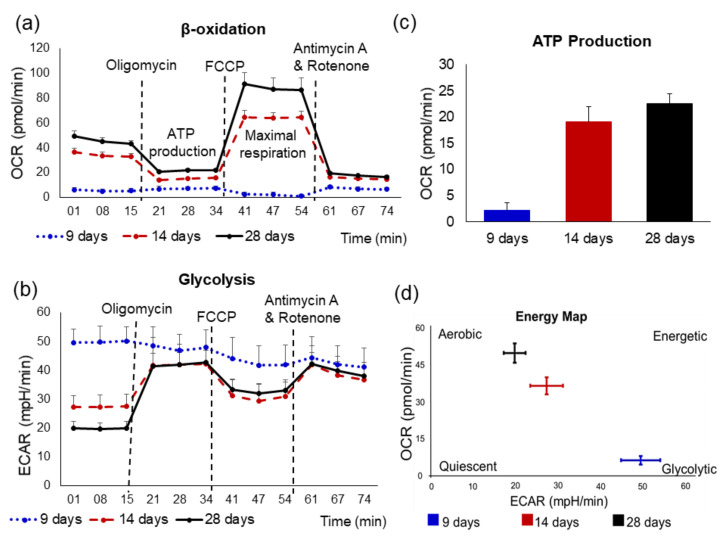
Analysis of CMs oxidative metabolism using a Seahorse XFe96 Analyzer to examine changes in metabolic profiles of hiPSC-derived-CMs following 9, 14, and 28 days of differentiation. The oxidative phosphorylation and glycolysis in CMs were measured through OCR (**a**) and ECAR (**b**), respectively. CM respiration was assayed under basal conditions and after addition of electron transport chain inhibitors, which were mitochondrial inhibitor oligomycin (2 μM), mitochondrial uncoupler carbonylcyanide p-trifluoromethoxyphenylhydrazone (FCCP; 0.3 μM) as well as a respiratory chain inhibitor antimycin A (0.5 μM) and rotenone (0.5 μM). (**c**) The averaged levels of ATP production. (**d**) Analysis of cell energy phenotype at 9 days hiPSC-derived-CMs were glycolysis-dependent and this is followed by a shift to β-oxidation metabolism at 14 and 28 day old CMs. These results indicate that a change in the cellular energy phenotype is accompanied by increasing the rate of O_2_ consumption and ATP synthesis to fulfill the needs of mature CMs.

**Figure 4 cells-10-03483-f004:**
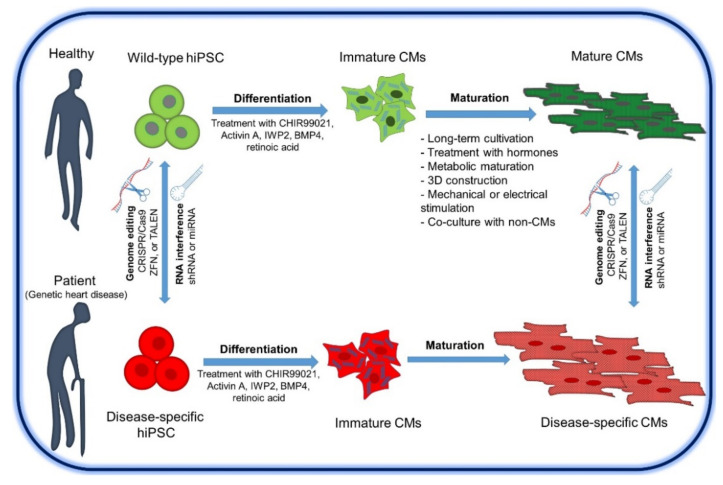
Summary of the strategies for generating hiPSC for heart disease modeling and protocols to improve the maturation of hiPSC-derived-CMs. Abbreviation: Clustered Regular Interspaced Short Palindromic Repeats-associated Cas9 nuclease, (CRISPR/Cas9); transcription activator-like effector nucleases (TALENs); Zinc finger nucleases, (ZFNs).

**Table 1 cells-10-03483-t001:** Summary of the differences between immature and mature CMs.

Parameter	Immature CMs	Mature CMs
Cell shape	Irregular	Rod-shaped
Cell area	480 ± 32 µm^2^	1716 ± 150 µm^2^
Sarcomere structure	Disorganized and less-developed	Organized and M-line developed
Sarcomere length	≈1.65 µm	≈1.81−2.3 µm
Cardiac troponin I	ssTnI	cTnI
Titin	N2BA isoform (N2B and N2A)	N2B
MHC	β-MHC ≤ α-MHC	β-MHC >> α-MHC
T-tubules	Deficient	Abundant
Mitochondria	Round and cristae are less mature	Elongated and more mature cristae (up to 40% cell volume)
Sarcoplasmic reticulum network	Underdeveloped	Well-developed
Metabolic	Glycolysis (80%)	Fatty acid β-oxidation 70–80%)
Nucleation	Mononuclear	Dinuclear (~35.3%)
Upstroke velocity	≈44 to 50 V/s	≈188.7 to 250 V/s
Resting membrane potential	≈ −50 to −60 mV	≈ −90 mV
Gap junction	Circumferentially	Periphery at the intercalated disc
Adrenergic receptors	≈β1 ≤ β2	≈70 β1:30 β2 and response to β-adrenergic stimulation
SERCA2a	Low expressed	Highly expressed
Contraction	Asynchronous	Synchronous

## Data Availability

Not applicable.

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
