# Peer review of "Human Induced Pluripotent Stem Cell as a Disease Modeling and Drug Development Platform—A Cardiac Perspective"

_cells, 2021, doi:10.3390/cells10123483_

Round 1
Reviewer 1 Report
Bekhite et al. compiled a comprehensive review of human iPSC-based heart disease modeling. The authors did an excellent job of introducing the iPSC technology as well as previous and current studies in the cardiac field. I only have three suggestions:
1. Other 3D strategies that are currently being used in the field include: engineered heart tissue/caridac organoid are missing. This section must be discussed by the authors.
2. The advantages of iPSC-CM in drug development were thoroughly highlighted by the authors. However, aside from toxicity studies, there are few excellent examples/references of drug development using iPSC-CMs.
3. A separate paragraph describing the limitation of iPSC-CMs could be useful for the general reader.
Author Response
Reviewer 1:
We are very happy to recognize that the reviewer is satisfied with the quality of our work and we thank the reviewer for his comments and suggestions
Minor comments:
- Comment: Other 3D strategies that are currently being used in the field include: engineered heart tissue/caridac organoid are missing. This section must be discussed by the authors.
Response: We thank the reviewer for this comment and additional information with a new citation for using human cardiac organoids and cardiac tissue engineering in modeling cardiovascular diseases, drug screening, and cardiac repair were provided in the revised version (Page 12, line 471-480 in the revised version).
- Comment: The advantages of iPSC-CM in drug development were thoroughly highlighted by the authors. However, aside from toxicity studies, there are few excellent examples/references of drug development using iPSC-CMs.
Response: We are thankful for this comment. Unfortunately, our description was not cover enough examples/references of drug development using iPSC-CMs. Therefore we added new examples with recent literature in using hiPSC-derived-CMs as models for potential drugs development and possible use of these compounds in the therapy of heart disease (Page 14, line 544-561 in the revised version).
- Comment: A separate paragraph describing the limitation of iPSC-CMs could be useful for the general reader.
Response: This is an interesting point and we agreed to the suggestion regarding describing the limitation of iPSC-CMs. As on page 11 section 9, consideration of hiPSC for use in modeling heart disease, we already mentioned some of these limitations of iPSC-CMs. In addition, we added new information with a citation for other limitations of iPSC-CMs to be resolved in the 12. Conclusions and challenges (Page 15, line 621-627 in the revised version). However, this very interesting point and one paragraph will be not enough to cover all limitation points in each section, for that, we believe writing a new review will be more useful for the general reader as you suggest.
Thank you again for all your suggestions and comments. We hope you appreciate the specific changes we have made in response to these comments and that overall you feel that the main arguments and contribution are now much stronger as a result.
Reviewer 2 Report
This is a well written review that would benefit a new investigator in the field, it covers from reprogramming to differentiation to maturation and disease modeling and drug development. I have major concerns on the abstract. 1) Though fully matured cardiomyocytes would be great for disease modeling and drug development (if one is studying adult-onset diseases), it is certainly not true for regenerative medicine as it was shown that adult cardiomyocytes do not engraft after transplantation (PMID 10402450); 2) Another concern is that though the authors did discuss the immaturity of the hiPSC-CMs, in line 19 to 22, please revise the sentence to reflect that the fact that hiPSC-CMs are immature.
3) for table 1, please illustrate more on sarcoplasmic reticulum, i.e. define "underdeveloped" and "well-developed";
4) Under the "8. strategies for generating hiPSC for heart disease modeling", from line 397 to 404, titin was mentioned twice;
5) please check that all the citations back up the statement, i.g. 122 and 273 does not seem to support the statement well in lines 427 to 431; this seems to be common case, please check the citations carefully;
Author Response
Reviewer 2:
Major concerns:
- 1Comment: Though fully matured cardiomyocytes would be great for disease modeling and drug development (if one is studying adult-onset diseases), it is certainly not true for regenerative medicine as it was shown that adult cardiomyocytes do not engraft after transplantation.
Response: We thank the reviewer for this comment. In the revised version we clarify that fully matured hiPSC-derived-CMs would be great for drug testing and studying adult-onset cardiac disease (Page 1, line 18 in the revised version). In addition, we added new information with a citation for other limitations of iPSC-CMs to be resolved in the 12. Conclusions and challenges (Page 15, line 621-627 in the revised version).
2. Comment: Another concern is that though the authors did discuss the immaturity of the hiPSC-CMs, in line 19 to 22, please revise the sentence to reflect that the fact that hiPSC-CMs are immature.
Response: This comment is very helpful to state more precise and we have revised the sentence to reflect that the fact that hiPSC-CMs are immature (Page 1, line 19-20 in the revised version).
3. Comment: for table 1, please illustrate more on sarcoplasmic reticulum, i.e. define "underdeveloped" and "well-developed";
Response: We included some more information to define the undeveloped sarcoplasmic reticulum (SR). That is, express lower SERCA and cardiac SR luminal auxiliary proteins (calsequestrin, junctin, and triadin), which form a protein complex associated with RyR2. (Page 6, line 241-251 in the revised version)
4. Comment: Under the "8. strategies for generating hiPSC for heart disease modeling", from line 397 to 404, titin was mentioned twice;
Response: Titin was mentioned first time to the most common mutated gene in familial DCM cases and for the second time to mentioned to hiPSC-derived-CMs have been established to study inherited DCM, including desmin (DES) (Page 11, line 414-418 in the revised version)
5. Comment: please check that all the citations back up the statement, i.g. 122 and 273 does not seem to support the statement well in lines 427 to 431; this seems to be common case, please check the citations carefully;
Response: Thank you for pointing to these issues. We have revised all citations carefully and removed the citation that does not support the statement.
Thank you again for all your suggestions and comments. We hope you appreciate the specific changes we have made in response to these comments and that overall you feel that the main arguments and contribution are now much stronger as a result.